# Proteomic Analysis of the Protective Effect of Early Heat Exposure against Chronic Heat Stress in Broilers

**DOI:** 10.3390/ani10122365

**Published:** 2020-12-10

**Authors:** Da Rae Kang, Kwan Seob Shim

**Affiliations:** 1Department of Animal Biotechnology, Jeonbuk National University, Jeonju 54896, Korea; kangdr92@gmail.com; 2Department of Agricultural Convergence Technology, Jeonbuk National University, Jeonju 54896, Korea

**Keywords:** heat stress, early heat exposure, proteomics, broiler

## Abstract

**Simple Summary:**

Heat stress affects the livestock industry, especially in poultry. Screening for metabolic changes after early and chronic heat exposure in poultry would be beneficial in resolving the production issues. In this study, we identified differentially expressed proteins that affected early heat exposure during chronic heat stress. Chronic heat stress affected 277 proteins, of which 95 differed in expression by early heat exposure. Differentially expressed proteins were related to actin metabolism and also involved in carbohydrate and carbon metabolism. According to our results, early heat exposed liver of broilers activates the different physiological mechanisms for protection from later heat stress.

**Abstract:**

The increasing trend of global warming has affected the livestock industry through the heat stress, especially in poultry. Therefore, a better understanding of the mechanisms of heat stress in poultry would be helpful for maintaining the poultry production. Three groups were designed to determine early heat stress effects during chronic heat stress: CC, raised at a comfortable temperature; CH, chronic heat exposure at 35 °C for 21–35 days continuously; and HH, early heat exposure at 40 °C for 24 h at 5 days old with 35 °C temperature for 21–35 days continuously. In this study, proteome analysis was carried out to identify differentially expressed proteins in the liver tissue of broilers under chronic and early heat exposure. There were eight differentially expressed proteins from early heat stress during chronic heat exposure, which were related to actin metabolism. According to KEGG (Kyoto encyclopedia of genes and genomes) analysis, the proteins involved in carbohydrate metabolism were expressed to promote the metabolism of carbohydrates under chronic heat stress. Early heat reduced the heat stress-induced expression changes of select proteins. Our study has shown that early heat exposure suggests that the liver of broilers has various physiological mechanisms for regulating homeostasis to aid heat resistance.

## 1. Introduction

The poultry industry is influenced by a number of factors, including the climate, environment, breeding, specific conditions, and heat stress. Poultry are particularly sensitive to high temperatures because they lack biological properties to release body heat; for example, they have no sweat glands and are covered with feathers [1]. Heat stress leads consistently to reductions in feed consumption, feed efficiency, growth rate, and reproduction, thereby severely affecting the health of animals and impairing their immune functions [2]. It can cause the dysfunction of organs and death, consequently leading to economic losses. Studies on broilers under high temperature stress have observed that their feed intake and growth rate are decreased [1], their respiration rates are increased [3], and the weights of different organs such as the thymus, spleen, and bursa of Fabricius are reduced [4]. The published studies on heat stress in poultry have mainly focused on the behavior, productivity, and blood biochemistry levels of the animals but not on the proteomic mechanisms of thermal reactions [1,4,5,6,7].

Early heat exposure increases the heat adaptability against heat stress later in poultry [5,6,7]. Interestingly, early heat exposure can be either pre- and post-hatching or both stages, which induces heat adaptability in the broiler. The heat treatment before hatching (16–18 days of incubation) and at both hatching stages combined (16–18 days of incubation and in 3-day-old chicks) did not induce heat resistance in the chicks at 42 days, whereas the post-hatching heat exposure (3-day-old chicks) contributed to heat resistance [6]. Early heat-treated birds exhibited significantly reduced mortality and improved feed efficiency [7], and responded similarly to birds adapted to heat stress [5]. These results indicated the protective effects of early heat exposure against heat stress later in life.

The techniques for studying genomics and proteomics are developing rapidly. Proteomic studies of heat stress effects have been mainly carried out on a number of plants and on animals such as dairy cows, beef cattle, pigs, and chickens [8,9,10,11]. In heat-stressed pigs, there was an increase in the expression of heat-shock proteins that protect cells by degrading denatured proteins; inflammatory cytokines, such as glucose-regulated protein 94 (GRP94) and serpin family A member 3 (SERPINA3); and glutathione peroxidase and glutathione S-transferase, which are related to immune responses [12]. Thus, numerous proteins and regulatory functions related to heat stress have been and are still being studied through proteomic analysis. Through studies of the whole proteome of various organs, the effects of heat stress can be confirmed at the overall metabolic level rather than at fragmented parts, making a detailed understanding of the heat effects possible.

Based on our previous results, it was hypothesized that early heat exposure in life may increase the resilience to heat stress at later stages of life [13]. To our knowledge, it is not well known about the effect of early heat exposure or chronic heat stress on overall protein changes. Those changed proteins through early heat exposure may lead to increasing their resistance to chronic heat stress. Hence, in this study, we conducted a proteomic analysis of broiler liver tissue under chronic heat stress and early heat exposure, to identify the effects of early heat exposure in poultry.

## 2. Materials and Methods

### 2.1. Animal Experiments

Experimental procedures for animals were approved by the Animal Ethics Committee of the Jeonbuk National University (CBNU2018-097), Republic of Korea. In total, 144 Ross 308 chickens (1-day-old) were purchased from Dongwoo Hatchery (Iksan, Korea). The chicks were weighed and randomly grouped into 3 groups of 48 chicks each without significant weight differences. Each group contained 4 replications and each replication obtained 12 chicks. All 1-day-old chicks were initially raised at 34 °C and the temperature was reduced by 2 °C weekly until it reached 24 °C. The humidity was maintained at 57 ± 3% throughout the experiment. The chicks had access to water and feed ad libitum. The following were the heat treatment groups: Group CC, raised at a suitable temperature without heat exposure; Group CH, chronic heat exposure at 35 °C for 24 h/day between the ages of 21 and 35 days; and Group HH, early heat exposure at 40 °C for 24 h at 5 days of age and chronic heat exposure at 35 °C for 24 h/day between the ages of 21 and 35 days (Figure 1). After the respective treatment, we calculated the body weight, feed intake, body weight gain, and feed conversion ratio (FCR) of the broiler. We demonstrated these growth performance-related data in our previous study [13]. Thereafter, twenty-seven birds (9 birds/group) were sacrificed, and the liver tissues were extracted, frozen with liquid nitrogen, and stored at −80 °C until analysis.

### 2.2. Protein Extraction and Digestion

A total of 27 samples were pooled to 9 samples (3 pooled samples/group) and extracted. An equal amount of liver tissue from each chicken was lysed in 1 mL of 8 M urea and protease inhibitor with vortex mixing for 30 s. Then, the mixture was sonicated for 3 min in an ice bath to homogenize the tissue. The homogenized tissue was then centrifuged at 14,000 rpm for 10 min, following which the supernatant was obtained for protein concentration measurement with the bicinchoninic acid assay. Then, 100 μg of protein was subjected to in-solution digestion. Thereafter, 100 µg of depleted protein in 100 mM Tris buffer (pH 3.0) in a total volume of 30 µL was first incubated with 6 M urea and 20 mM dithiothreitol at 56 °C for 30 min and then alkylated with fresh 50 mM iodoacetamide in 100 mM Tris buffer (pH 8.0) for 30 min in the dark at ambient temperature (24 °C). The reaction was quenched with 100 mM Tris buffer (pH 8.0) and the protein was enzymatically digested overnight at 37 °C in a trypsin/LysC mix (1:50, enzyme:substrate). The reaction was quenched with formic acid and the peptides were desalted on an Oasis HLB column (Waters Corporation, Milford, MA, USA). Finally, the peptides obtained were dried using a speed vac.

### 2.3. Liquid Chromatography-Tandem Mass Spectrometry Analysis and Data Analysis

The peptides were separated by liquid chromatography (LC) on an EASY-nLC 1000 system (Thermo Fisher Scientific, Rockford, IL, USA) equipped with a C18 column (2 µm particle size, 50 μm ID × 15 cm length; Thermo Fisher Scientific, California, CA, USA), using mobile phase A (0.1% formic acid in water) and B (0.1% formic acid in 100% acetonitrile) at a flow rate of 300 nL/min. The gradient profile was set as follows: 5–40% B in 45 min and 40–80% B in 2 min. Mass spectrometry (MS) analysis was performed using a Q Exactive mass spectrometer (Thermo Fisher Scientific, Bremen, Germany) with the spray voltage set at 2.3 kV. MS spectra were collected at a resolution of 70,000 at *m*/*z* 200 (350–2000 *m*/*z* mass range), followed by data-dependent higher-energy collisional dissociation (HCD) MS/MS spectra (at a resolution of 17,500 and collision energy of 25%) of the 20 most abundant ions. A dynamic exclusion time of 30 s was used.

For the identification of the chicken liver protein, the raw files were searched against the UniProt Chicken database using the Percolator node in Proteome Discoverer (v1.4, Bremen, Germany). Oxidation was chosen as the dynamic modification and carbamidomethyl as the static modification. The parent ion mass error was set to ±10 ppm and the fragment ion mass error to ±0.02 Da. Peptides with full tryptic cleavage specificity were searched, with two missed cleavages allowed. The search parameters used were a precursor tolerance of 10 ppm and a fragment ion tolerance of 0.02 Da. Scaffold Q+S was used for the label-free quantitation of the proteins. All identified proteins were normalized by protein size and total spectrum count number, and the normalized data were filtered with a confidence level (CI) of >95%. For further filtering, the false discovery rate was set to 0.05 and the count number to >0 for 2 out of 2 replicates.

### 2.4. Gene Ontology Enrichment Analysis

The differentially expressed proteins were clustered using the Database for Annotation, Visualization and Integrated Discovery bioinformatics resources (DAVID v6.8, https://david.ncifcrf.gov/) with the Gene Ontology (GO) database, and the probability value was corrected with the Bonferroni method. The classifications from the Gallus gallus database were applied. Pathway enrichment analysis was carried out using the WEB-based Gene SeT AnaLysis Toolkit (WebGestalt, http://www.webgestalt.org/) with the Kyoto Encyclopedia of Genes and Genomes (KEGG) database. The classifications were again from the Gallus gallus database. The probability value was corrected with the Benjamini–Hochberg method.

### 2.5. Validation of Proteins by Their Gene Expression

Analyzing the gene expression level is to confirm the protein expression level. Total RNA was extracted from the liver tissue using an RNA extraction kit (Bioneer, Daejeon, Korea) according to the manufacturer’s instructions. The RNA concentration and purity were measured using the μDrop Plate (NanoDrop, Thermo Fisher Scientific, Delaware, DE, USA). cDNA was synthesized from 1 µg of total RNA using the AccuPower RocketScript Cycle RT PreMix (dT20) (Bioneer, Daejeon, Korea). The randomly selected gene primers were designed using *gallus gallus* genes with Primer 3 software (v.0.4.0, https://bioinfo.ut.ee/primer3-0.4.0/) and are shown in Appendix A. The reverse transcription quantitative polymerase chain reaction (RT-qPCR) was conducted using the SsoFast EvaGreen Supermix (Bio-Rad, Hercules, CA, USA) on a CFX96 real-time PCR detection system (Bio-Rad, California, USA). The RT-qPCR thermal cycle conditions were as follows: 95 °C for 5 min, followed by 40 cycles of 95 °C for 5 and 60 °C for 30 s. The relative gene expression levels were calculated with the 2^−ΔΔCt^ method [14] and normalized against the level of glyceraldehyde 3-phosphate dehydrogenase (GAPDH).

### 2.6. Statistical Analysis

The experimental data for mRNA expression was analyzed with the SAS 9.4 program (North Carolina, USA) and are expressed as the mean ±SE. Differences were analyzed using analysis of variance, and statistical differences among groups were analyzed with Duncan’s multiple-range test. Statistical significance was set at *p* < 0.05.

## 3. Results

### 3.1. Differentially Expressed Proteins in Response to Chronic Heat Stress and Early Heat Exposure

In total, 277 proteins (132 downregulated and 145 upregulated) were differentially expressed between the control and CH groups (Appendix A). However, the chronic heat affect was reduced by early heat exposure for 95 (42 downregulated and 53 upregulated by chronic heat stress) of the 277 proteins (Appendix A). The heat map and graph of the expression patterns of the differentially expressed proteins in the various groups are shown in Figure 2. Of the 95 proteins that were positively affected by the early heat exposure, the putative interferon-stimulated gene 12 (ISG12) protein was the most highly expressed in response to chronic heat stress. Therefore, the role of this protein against viral infections and in regulating the immune system may be regulated by early heat exposure.

To further compare the differences among the CC, CH, and HH groups, the functions of the differentially expressed proteins were categorized to the GO resource, in which the three main functional categories are biological process, cellular component, and molecular function. Using the 95 proteins that were positively affected by early heat treatment, we obtained the top 10 GO terms for the three main categories (Appendix A and Figure 3). In the biological process, genes were associated with the small molecule metabolic process (13 genes), actin filament-based process (7 genes), actin cytoskeleton organization (7 genes), and cofactor metabolic process (7 genes). In the cellular component, actomyosin had the highest significance among the GO terms, with the 4 genes *actinin alpha 1* (*ACTN1*), *fermitin family homolog 2* (*FERMT2*), *LIM domain and actin binding 1* (*LIMA1*), and *filamin B* (*FLNB*) being associated with the actomyosin, stress fiber, contractile actin filament bundle, actin filament bundle, focal adhesion, cell-substrate adherens junction, adherens junction, and actin cytoskeleton terms. In the molecular function, the genes were strongly associated with actin filament binding, with the 5 genes *ACTN1*, *FERMT2*, *LIMA1*, *myosin heavy chain 9* (*MYH9*), *myosin 1B* (*MYO1B*) and *WD repeat-containing protein 1* (*WDR1*) being enriched in the actin filament binding, actin binding, cytoskeletal protein binding, and protein-containing complex binding terms. According to GO analysis, differentially expressed protein cording genes by early heat were highly associated with actin metabolism.

On the basis of the KEGG pathway, we identified 13 significant pathways among the CC, CH, and HH groups (Table 1). Among them, the pathways involved in carbohydrate metabolism including those of glycolysis/gluconeogenesis, citrate cycle, pentose and glucoronate interconversions, fructose and mannose metabolism, starch and sucrose metabolism, glyoxylate and dicarboxylate metabolism, and propanoate metabolism were the most common.

### 3.2. Gene Expression Analysis for Validation of the Abundant Protein

Gene expression levels were analyzed to verify proteomic analysis. We analyzed the mRNA expression levels of glutamic-oxaloacetic transaminase 1 (GOT1), trifunctional purine biosynthetic protein adenosine-3 (GART), *glutathione S-transferase theta 1-like* (GSTT1L), *2-oxoglutarate dehydrogenase-like, mitochondrial* (OGDHL), *UDP glucuronosyltransferase family 2 member A1* (UGT2A1), arginyl-tRNA synthetase, cytoplasmic (RARS), cytochrome P450 2D6 (CYP2D6), *isoleucyl-tRNA synthetase* (IARS), and glutathione synthetase (GSS) (Figure 4). Those genes were randomly selected from genes encoding proteins related to the KEGG pathway. The OGDHL and UGT2A1 expression levels were significantly higher in the CH and HH groups than in the CC group (*p* < 0.05), where the pattern was similar to that of their protein expression levels; that is, the highest in the CH group, lowest in the CC group, and moderate in the HH group. The GOT1, GART, GSTT1L, RARS, CYP2D6, and GSS expression levels were significantly higher in the CH group than in the CC and HH groups (*p* < 0.05), whereas IARS expression was significantly lower in the CH group (*p* < 0.05). GOT1, GART, and GSTT1L showed the same gene and protein expression patterns (showed in Appendix A), whereas CYP2D6, IARS, and GSS showed high gene and low protein (or low gene and high protein) expression patterns.

## 4. Discussion

Given that protein metabolism in animals can be altered by heat, researchers have studied the heat-induced proteomic responses in various animals, such as cattle [9], pigs [11], and birds [8,10]. For example, Victoria et al. [11] reported that heat stress affects carbohydrate metabolism in pigs. Our differentially expressed proteins were consistent with the results of these reports, in that proteins in pathways of carbohydrate metabolism—such as those in glycolysis/gluconeogenesis (acetyl-CoA synthetase 2-like, mitochondrial isoform X1 (ACSS1L), glucose 6-phosphate isomerase (GPI), hexokinase domain-containing 1 (HKDC1)), the citrate cycle (OGDHL), starch and sucrose metabolism (GPI, HKDC1, UGT2A1), and propanoate metabolism (ACSS1L, propionyl-CoA carboxylase subunit alpha (PCCA))—were upregulated in the liver tissue of the CH chickens compared with that of the CC and HH chickens. The overall metabolic process is shown in Figure 5.

It has been found that enzyme activity increases exponentially at high temperatures (above 55 °C) [15]. GPI protein has been studied as a neurotrophic factor for promoting the survival of skeletal and sensory neurons and inducing immunoglobulin secretion, and as a tumor-secreting cytokine that plays a role in tumor angiogenesis and metastasis and cell migration, proliferation, and apoptosis [16]. Therefore, the upregulation of both HKDC1 and GPI by chronic heat stress would promote glycolysis and activate pathways to obtain energy from glucose in the body. However, such overexpression may induce cell damage as well. Sorbitol dehydrogenase (SORD), which converts sorbitol to fructose in the polyol pathway, is closely related to various diabetic complications (viz., neuropathy, retinopathy, cataracts, and nephropathy) [17]. The decreased expression of SORD causes an excessive accumulation of sorbitol, leading to osmotic damage to the retinal endothelial cells and pericytes through oxidative-nitridation stress and activation of the protein kinase C pathways, with resultant inflammation and growth factor imbalance [18]. However, the SORD reduction in the CH group was recovered in the HH group, indicating that early heat exposure plays a protective role against heat stress in the liver cells.

In the tricarboxylic acid (TCA) cycle, the main precursor acetyl-CoA is essential for energy generation toward the mitochondria [19]. In broilers, ACSS1L catalyzes the synthesis of acetyl-CoA for use in glycolysis. In general, broilers subjected to heat stress have reduced feed intake, and thus the expression of ACSS1L was increased in that group owing to the insufficient nutrients for metabolism. Citrate synthase (CS) catalyzes the condensation reaction to form citrate from oxaloacetate and acetyl-CoA, which are the first steps in the TCA cycle. It is also used as an enzymatic marker of intact mitochondria [20]. From the results of our study, we can postulate that heat stress causes damage to the mitochondria as a result of the downregulation of CS, but early heat exposure can regulate the CS expression level. OGDHL, which is located in the mitochondria, is also a TCA cycle-related enzyme and indirectly responsible for the induction of apoptosis. These results suggest that overexpressed OGDHL plays an important role in inducing ROS-mediated apoptosis. In our study, chronic heat stress induced OGDHL expression, which suggests that it may induce ROS production followed by cell damage. Those changes in proteins related to carbohydrate metabolism may have an effect on feed intake due to heat stress [13].

In general, heat stress increases energy consumption for panting to dissipate heat, maintaining homeostasis, and protecting the cells [21]. In a study on the fatty acid levels in broilers under heat stress, the plasma concentration of non-esterified fatty acids was reduced in the heat-stressed animals, and better absorption and storage of triglycerides in the intestine or liver were observed [22]. Interestingly, liver can regulate intestinal integrity. A previous study reported that chronic liver disease altered the intestinal homeostasis [23]. It has been reported that acute and chronic heat stress altered the liver function and HSPs gene expression in the broiler’s small intestine [24,25]. This is because the heat emitted from fat metabolism is higher than that generated through carbohydrate metabolism. Similarly, in heat-stressed dairy cows, energy was preferentially supplied by the carbohydrates to reduce the body’s own heat generation [26]. However, in our study, chronic heat stress, which increases the requirement of a considerable amount of energy, increased the expression of enzymes of the acyl-CoA dehydrogenase (ACAD) family (i.e., ACAD11 and ACAD9) to obtain energy through fat. This confirms that a considerable amount of energy is required, along with a reduction in feed intake, in response to chronic heat stress. Although heat stress significantly reduced the nutrient (and thus potential energy) intake by the broilers, a large amount of energy was needed from metabolism to cope with the stress. Thus, the expression of carbohydrate metabolism-related proteins changed.

In general, since the mRNA is finally translated into a protein, the expression of the gene encoding the protein is analyzed to verify the protein expression. Therefore, we also analyzed the expression of coding genes of each protein. Some of the genes (*GOT, GSTT1L, OGDHL, UGT2A1, RARS*) we analyzed were the same pattern as each protein expression, but some (*GART, CYP2D6, GSS, IARS*) were not. These difference in gene and protein expression is probably because of changing the levels of transcription, mRNA processing, and the translation to protein due to heat stress [27,28,29].

Finally, the feasibility of the early heat exposure method for reducing heat stress in broilers was studied, as this method has been reported to have a protective effect against heat stress. Although there are published studies on the effects of heat stress on cell growth, heat-shock protein expression, hormones, and blood parameters, to the best of our knowledge, no research has been conducted on the mechanisms related to the early heat reactions. Thus, in this study, we carried out a proteomic analysis to confirm which proteins/processes are regulated by early heat exposure under chronic heat stress. The results showed that chronic heat stress affected the expression of proteins involved in carbohydrate metabolism and carbon metabolism, which generally result in less heat production than lipid metabolism for energy production. If the expression of these proteins increases, cell damage may increase as well. However, the early heat treatment recovered the proteins that were induced (or reduced) by chronic heat stress, indicating that early heat exposure maintains homeostasis and inhibits biological damage in response to heat stress. The knowledge about these physiological changes would be useful for the development of programs for breeding animals with high heat resistance. Moreover, this early heat exposure method and the results of our study may be helpful in the study of heat stress in animals for improving animal welfare.

## Figures and Tables

**Figure 1 animals-10-02365-f001:**
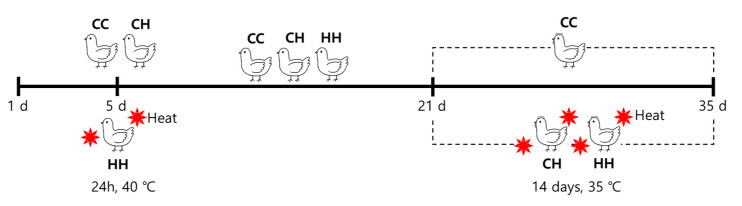
Schematized schedule of heat exposure condition. CC: raised at a suitable temperature without heat exposure. CH: chronic heat exposed group. HH: early and chronic heat exposed group.

**Figure 2 animals-10-02365-f002:**
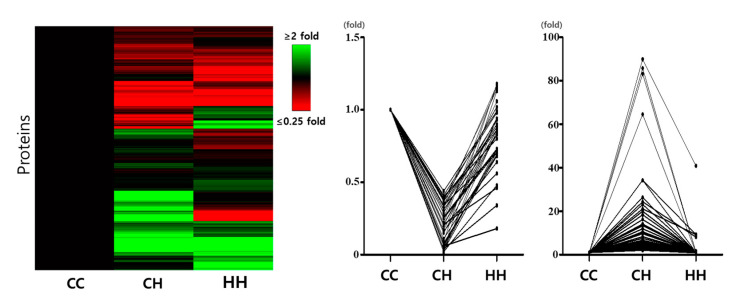
Heat map using differentially expressed proteins and expressed patterns of positive effected proteins by early heat. Each value is a fold change compared to the CC. CC: raised at a suitable temperature without heat exposure, CH: chronic heat exposed group, HH: early and chronic heat exposed group.

**Figure 3 animals-10-02365-f003:**
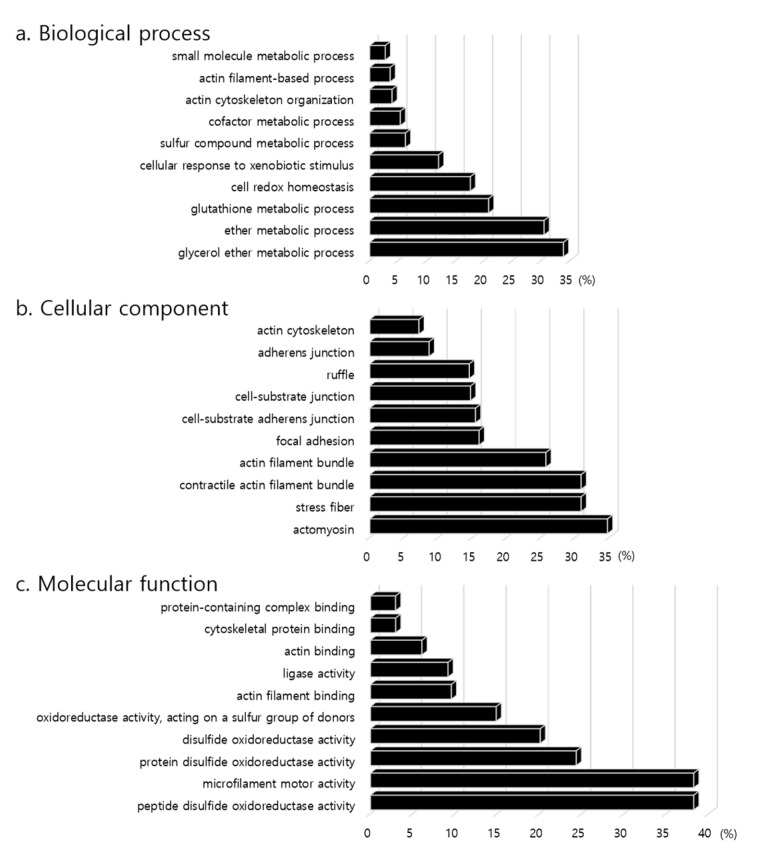
Gene ontology terms of positive effected proteins of differentially expressed proteins by early heat exposure. The *x*-axis unit of percentage (%) is ‘the number of genes that included in the GO term/the total number of genes in GO term’.

**Figure 4 animals-10-02365-f004:**
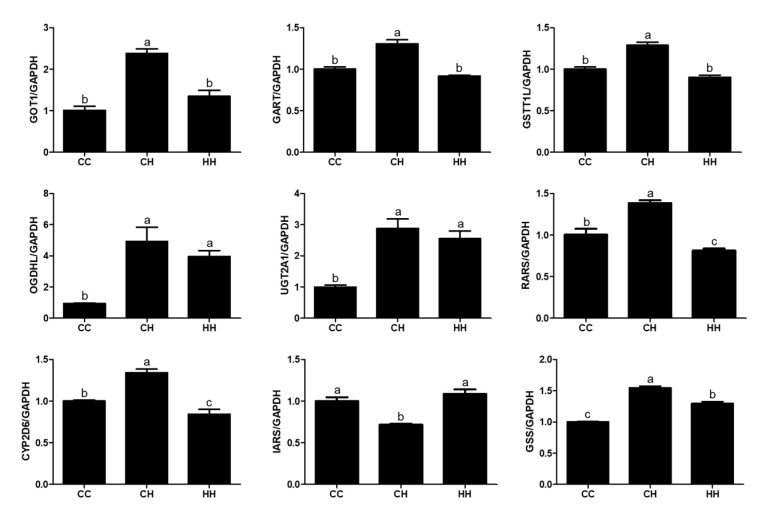
Gene expression in the liver tissue. The values were expressed with fold-change compared to CC. CC: raised at a suitable temperature without heat exposure; CH, chronic heat exposed group; HH, early and chronic heat exposed group. ^a–c^ Different superscript letters are significantly different (*p* < 0.05).

**Figure 5 animals-10-02365-f005:**
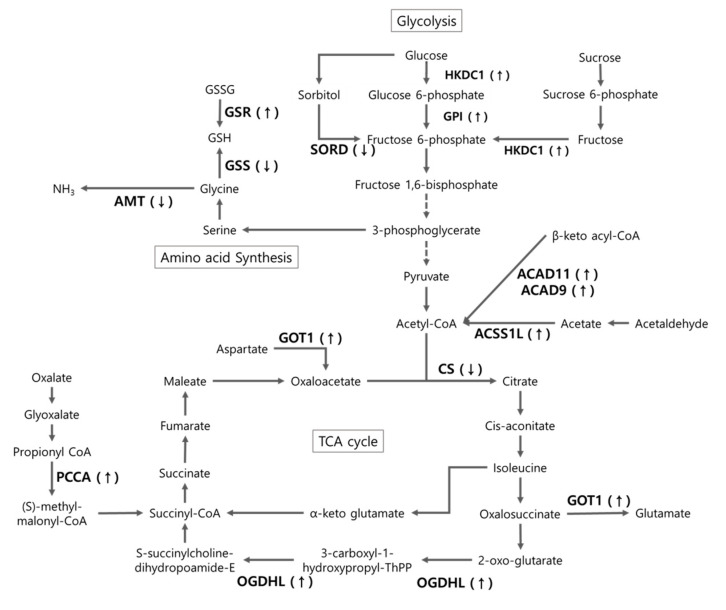
Metabolic schematic of differentially expressed proteins. (↑) or (↓) sign showed significantly up or downregulated in CH group compared to CC and/or HH.

**Table 1 animals-10-02365-t001:** List of KEGG pathways.

Term ID	Term	No.	Genes	*p*-Value
Carbohydrate metabolism				
00010	Glycolysis/Gluconeogenesis	3	*ACSS1L, GPI, HKDC1*	0.02
00020	Citrate cycle (TCA cycle)	2	*CS, OGDHL*	0.04
00040	Pentose and glucoronate interconversions	2	*SORD, UGT2A1*	0.02
00051	Fructose and mannose metabolism	2	*HKDC1, SORD*	0.04
00500	Starch and sucrose metabolism	3	*GPI, HKDC1, UGT2A1*	0.01
00630	Glyoxylate and dicarboxylate metabolism	4	*AMT, CS, PCCA, ACSS1L*	0.004
00640	Propanoate metabolism	2	*ACSS1L, PCCA*	0.04
Metabolism of cofactors and vitamins				
00670	One carbon pool by folate	2	*AMT, GART*	0.01
00860	Porphyrin and chlorophyll metabolism	2	*UGT2A1, UROD*	0.03
Xenobiotics biodegradation and metabolism				
00982	Drug metabolism (cytochrome P450)	2	*GSTAL3, GSTT1L*	0.04
00983	Drug metabolism (other enzymes)	3	*LOC769704, GSTAL3, GSTT1L*	0.02
Translation				
00970	Aminoacyl-tRNA biosynthesis	4	IARS, QARS, RARS, TARS	0.001
Metabolism of other amino acids				
00480	Glutathione metabolism	3	*GSR, GSS, GSTT1L*	0.01

*ACSSlL: acetyl-CoA synthetase 2-like, mitochondrial isoform X1, GPI: glucose 6-phosphate isomerase, HKDC1: hexokinase domain-containing 1, CS: citrate synthase, OGDHL: 2-oxoglutarate dehydrogenase-like, mitochondrial, SORD: sorbitol dehydrogenase, UGT2A1: UDP glucuronosyltransferase family 2 member A1, AMT: aminomethyltransferase, mitochondrial, PCCA: propionyl-CoA carboxylase subunit alpha, GART: trifunctional purine biosynthetic protein adenosine-3, UROD: uroporphyrinogen decarboxylase, GSTAL3: glutathione S-transferase 3, GSTT1L: glutathione transferase, IARS: isoleucyl-tRNA synthetase. QARS: glutaminyl-tRNA synthetase, RARS: arginyl-tRNA synthetase, TARS: threonyl-tRNA synthetase, GSR: glutathione reductase.*

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
