# Peer review of "Proteomic Analysis of the Protective Effect of Early Heat Exposure against Chronic Heat Stress in Broilers"

_animals, 2020, doi:10.3390/ani10122365_

Round 1
Reviewer 1 Report
General remarks:
The manuscript addresses the effects of heat stress in broilers, a subject of global relevance. The experimental design follows the hypothesis that exposure of poultry chicks in an early stage of life (here day 5 in the HH group) may condition the animals to heat stress experienced in later stages of life (21-35 days of life, CH and HH). Results were compared to a reference group (CC) kept at an ambient temperature regimen. For the interpretation of the results and full validation of the significance of the presented results, the authors are invited to answer/comment on the following questions:
Why are no clinical parameters, such as feed intake and weight gain are reported? Interrupted or a measurable decline in feed intake during the heat stress period could readily explain various of the measured changes, as for example alteration in the carbohydrate metabolism. Please add the missing data and include this aspect also in the discussion section
Please explain the rationale for the selection of the test parameters. In research addressing the effects of heat stress, gene expression and their transcription of typical target genes, such as heat shock proteins and heat shock factor seem t be mandatory. Why were this genes not included in the analyses? Moreover, heat stress is well known to affect the intestinal barrier in poultry. This triggers again the question, why related genomic and proteomic data are not reported, as these changes in the intestines have a significant effect on liver functions. At least in the discussion, this aspect needs to be clearly addressed.
Specific comments:
The Simple Summary and the Abstract need considerable rephrasing, to clearly guide the reader into the subject and summarize the main results. In this context, the authors need to consider also minor editorials such as the wording “thermal reactions”, which is incorrect in this context an need to be replaced by phrases such as “reactions following exposure to heat stress”, “following exposure to high temperature “or similar phrases.
Figure 2: in the graphic display the scaling of the Y axis is not given, please add.
Figure 4: In this composite figure, the Y axis uses different scaling. Using a harmonized (constant) sailing would improve very much the readability of the table and allowing the reader to identify at one glance significant changes (only changes by a factor >2 are considered to be of biological significance, while minute statistical differences (<1 fold) are in most cases biologically insignificant and without consequences for animal welfare and productivity).
The annotation of various genes, particularly those related to xenobiotic metabolism, refer to the corresponding human genes (like CYP2D6, UGT 2A). Please confirm that specific primers for poultry were used and change the annotations in the text.
The discussion section seems to be overcomplete, in summarizing all potential functions of the measured genes in different organisms. A comprehensive discussion of the effects on poultry metabolism, animal health and welfare would be preferred.
Author Response
To reviewer.
Thank you for the time and effort you and each of the reviewers have dedicated to providing insightful feedback on ways to strengthen our paper. Thus, it is with great pleasure that we resubmit our article for further consideration. We have incorporated changes that reflect the detailed suggestions you have graciously provided. We also hope that our edits and the responses we provide below satisfactorily address all the issues and concerns you and the reviewers have noted.
Reviewer 1.
General remarks:
The manuscript addresses the effects of heat stress in broilers, a subject of global relevance. The experimental design follows the hypothesis that exposure of poultry chicks in an early stage of life (here day 5 in the HH group) may condition the animals to heat stress experienced in later stages of life (21-35 days of life, CH and HH). Results were compared to a reference group (CC) kept at an ambient temperature regimen. For the interpretation of the results and full validation of the significance of the presented results, the authors are invited to answer/comment on the following questions:
Q. Why are no clinical parameters, such as feed intake and weight gain are reported? Interrupted or a measurable decline in feed intake during the heat stress period could readily explain various of the measured changes, as for example alteration in the carbohydrate metabolism. Please add the missing data and include this aspect also in the discussion section
Answer: We agree with your perspective. In the previous studies, we analyzed the effects of early heat exposure in broilers under chronic heat stress through the same feeding method in feed intake, growth performance, GOT, GPT, and HSPs expression (Kang et al., 2019). As a result, it was derived that early heat exposure affects FCR and HSPs expression, and showed a positive effect on chronic heat stress. As a further study of the previous study, this study attempted to investigate the effect of early heat exposure on proteomic approach under chronic heat stress. In this study, we focused on identifying the metabolic mechanisms affected by heat stress and early heat by analyzing the patterns of changes in proteins and other basic data (feed intake, HSPs, blood parameters etc.) were not analyzed except body weight. Therefore, additional explanations have been prepared using the results of previous studies as references.
- Kang, D., Park, J., & Shim, K. (2019). Heat Treatment at an Early Age Has Effects on the Resistance to Chronic Heat Stress on Broilers. Animals, 9(12), 1022.
Q. Please explain the rationale for the selection of the test parameters. In research addressing the effects of heat stress, gene expression and their transcription of typical target genes, such as heat shock proteins and heat shock factor seem t be mandatory. Why were this genes not included in the analyses? Moreover, heat stress is well known to affect the intestinal barrier in poultry. This triggers again the question, why related genomic and proteomic data are not reported, as these changes in the intestines have a significant effect on liver functions. At least in the discussion, this aspect needs to be clearly addressed.
Answer: We also agree with your comment. And, similar to the above answer, this study is further study, and we tried to investigate the effect of early heat exposure on liver proteome in broiler chickens under chronic heat stress. Therefore, liver tissue was extracted to analyze the protein expression level, and the gene expression was analyzed for confirm the protein expression which measured by LC-MS/MS. We identify to 277 proteins which are responsible for the response the heat stress. Among these proteins we selected GOT1, GART, GSTT1L, OGDHL, UGT2A1, RARS, CYP2D6, IARS, and GSS gene for confirmation. In our previous study demonstrated that HSP70 and HSP27 expression were higher, and HSF3 expression was decreased in chronic heat stress (Kang et al., 2019). And liver can regulate intestinal integrity. Previous study reported that chronic liver disease altered the intestinal homeostasis (Schnabl, 2012). It has been reported that acute and chronic heat stress altered the liver function and HSPs gene expression in broiler small intestine (Shiddiqui et al., 2020a,b). And we added this information within the discussion part.
- Kang, D., Park, J., & Shim, K. (2019). Heat Treatment at an Early Age Has Effects on the Resistance to Chronic Heat Stress on Broilers. Animals, 9(12), 1022.
- Schnabl, P. (2012). The international transmission of bank liquidity shocks: Evidence from an emerging market. The Journal of Finance, 67(3), 897-932.
- Siddiqui, S. H., Kang, D., Park, J., Khan, M., & Shim, K. (2020). Chronic heat stress regulates the relation between heat shock protein and immunity in broiler small intestine. Scientific reports, 10(1), 1-11.
- Hasan Siddiqui, S., Kang, D., Park, J., Choi, H. W., & Shim, K. (2020). Acute Heat Stress Induces the Differential Expression of Heat Shock Proteins in Different Sections of the Small Intestine of Chickens Based on Exposure Duration. Animals, 10(7), 1234.
Specific comments:
The Simple Summary and the Abstract need considerable rephrasing, to clearly guide the reader into the subject and summarize the main results. In this context, the authors need to consider also minor editorials such as the wording “thermal reactions”, which is incorrect in this context an need to be replaced by phrases such as “reactions following exposure to heat stress”, “following exposure to high temperature “or similar phrases.
Answer: We agree with your comments, so we have modified ‘Simple summary’ and ‘abstract’ according to your suggestion.
Figure 2: in the graphic display the scaling of the Y axis is not given, please add.
Answer: Thank you for checking the details, we have added the scale value of the Y axis of the graphs.
Figure 4: In this composite figure, the Y axis uses different scaling. Using a harmonized (constant) sailing would improve very much the readability of the table and allowing the reader to identify at one glance significant changes (only changes by a factor >2 are considered to be of biological significance, while minute statistical differences (<1 fold) are in most cases biologically insignificant and without consequences for animal welfare and productivity).
Answer: We agree that the biological consideration if there is ‘>2 fold’ as your opinion. So we selected proteins with a difference of more than 2 times in protein expression (you can see in Figure 2) and analyzed their functions. However, Figure 4 is the data to verify the expression level of the protein by their coding genes using RT-PCR.
The annotation of various genes, particularly those related to xenobiotic metabolism, refer to the corresponding human genes (like CYP2D6, UGT 2A). Please confirm that specific primers for poultry were used and change the annotations in the text.
Answer: As you said, the CYP2D6 and UGT2A primer may reveal human gene. However we confirmed for the primers we designed through NCBI (primer-BLAST), and found that the primers correspond to poultry gene. Also add annotations in the text about ‘gallus gallus gene’ in materials and methods. For confirmation we attached NCBI file. And also you can check the ID for making primers in Supplementary Table 1.
The discussion section seems to be overcomplete, in summarizing all potential functions of the measured genes in different organisms. A comprehensive discussion of the effects on poultry metabolism, animal health and welfare would be preferred.
Answer: We have reflected this comment by modification the discussion section according to your suggestion.

Reviewer 2 Report
I recommend major revision. Authors have presented the influence of heat stress on expressed proteins in the liver tissue of broilers under chronic and early heat exposure. To this purpose the proteome analysis was performed. It very interesting paper, written with scientific accuracy. However, Authors should correct some points before the acceptance: 1. “Previous studies investigated that the early heat exposure has effects on late heat stress, but there usually analyzed about growth performances, serum parameters, heat shock proteins or other factors.” There should be given a reference. 2. “Our results suggest that proteins can be changed through early heat exposure to increase their resistance to chronic heat stress.” This sentence should be rewritten and put before the aim as a hypothesis. 3. “Ross” ? what kind of Ross? 4. How were chicks chosen to euthanasia among replications? 5. “the thermal stress of broilers has emerged as an economically viable method compared with other countermeasures” There a reference should be given. 6. “Although the specific biological functions of HKDC1 are still unclear, it has been proposed to play an especially greater role in glucose metabolism when the fetus needs to be supplied with sufficient nutrients during pregnancy” How it relates to chicks? 7. “Moreover, this early heat exposure method and the results of our study may be helpful in reducing heat stress in animals, thereby improving animal welfare in general.” It is over interpretation. There are needed more studies.
Author Response
To reviewer.
Thank you for the time and effort you and each of the reviewers have dedicated to providing insightful feedback on ways to strengthen our paper. Thus, it is with great pleasure that we resubmit our article for further consideration. We have incorporated changes that reflect the detailed suggestions you have graciously provided. We also hope that our edits and the responses we provide below satisfactorily address all the issues and concerns you and the reviewers have noted.
Reviewer 2
Comments and Suggestions for Authors
I recommend major revision. Authors have presented the influence of heat stress on expressed proteins in the liver tissue of broilers under chronic and early heat exposure. To this purpose the proteome analysis was performed. It very interesting paper, written with scientific accuracy. However, Authors should correct some points before the acceptance:
- “Previous studies investigated that the early heat exposure has effects on late heat stress, but there usually analyzed about growth performances, serum parameters, heat shock proteins or other factors.” There should be given a reference.
Answer: We agree with you. So we have added the reference about this sentence according to your suggestion.
- “Our results suggest that proteins can be changed through early heat exposure to increase their resistance to chronic heat stress.” This sentence should be rewritten and put before the aim as a hypothesis.
Answer: Thank you for your suggestion. We have rewritten that sentence to ‘Those changed proteins through early heat exposure may be lead to increase their resistance to chronic heat stress.’, and moved it to other location according to your suggestion.
- “Ross”? what kind of Ross?
Answer: We used ‘Ross 308’ in this study and we have added the information about species in materials and methods.
- How were chicks chosen to euthanasia among replications?
Answer: You have raised and important question. After the feeding and heat exposure experiment, nine chicks were randomly selected from each group.
- “the thermal stress of broilers has emerged as an economically viable method compared with other countermeasures” There a reference should be given.
Answer: We agree with your comment, however, we removed that sentence in the revised manuscript according to ‘Reviewer 1’ that said the discussion section seems to be overcomplete.
- “Although the specific biological functions of HKDC1 are still unclear, it has been proposed to play an especially greater role in glucose metabolism when the fetus needs to be supplied with sufficient nutrients during pregnancy” How it relates to chicks?
Answer: We agree with you about the sentence, that the sentence is about the role HKDC1 plays in mammals so we have removed that sentence. This is because our main species is poultry, which is different from mammals.
- “Moreover, this early heat exposure method and the results of our study may be helpful in reducing heat stress in animals, thereby improving animal welfare in general.” It is over interpretation. There are needed more studies.
Answer: We agree with your suggestion. We modified that sentence according to your suggestion of an over interpretation.
Round 2
Reviewer 1 Report
General concerns
The authors have answered to all questions addressed by this referee but failed to amend the manuscript accordingly (only very minor editorial changes were made). The introduction does not reflect the outcome of previous study, (See Kang, D.; Park, J.; Shim, K. Heat Treatment at an Early Age Has Effects on the Resistance to Chronic Heat Stress on Broilers. Animals, 2019, 9, 1022 – Journal missing in the ref list).
A major concern is the Discussion Section presenting a (text-book based) description of the genes/proteins under consideration but fails to establish a clear link to the effect of heat stress in poultry. Typical examples for the no-critical selection of literature information on metabolic pathways are the citation of references 9, 12, 16, 17, 18, 19, 21, 22, 24, 27, 29, 31, 32 which cover enterally different fields and disease condition.
It is strongly recommended that the authors discuss these results, with a competent colleague with broad insight into poultry physiology and pathophysiology and link the current results to the previous manuscript (Kang, D.; Park, J.; Shim, K. Heat Treatment at an Early Age Has Effects on the Resistance to Chronic Heat Stress on Broilers. 2019, 9, 1022). .
Specific comments repeated:
The Short abstract is still inadequate as it does to provide easy understandable information about the content of the manuscript.
Th authors should consider phrases like: based on our previous results, it was hypothesized that heat stress early in life may increase the resilience to heat stress at later stages of life. Hence in the current series of experiments chickens were exposed….
The recommended editorial changes in the figures were not implemented.
The recommendation to present a clear link with clinical data (previous experiments) was not considered (only a statement was given in the Rebuttal letter).
Gene expression differences were presented in percent (without standard deviation. see figure 3) while the general procedure is to present x-fold changes. As none of the changes apparently resulted in >2 fold differences in the level of expression, it seems surprising that major changes in protein expression levels were observed. This apparent discrepancy should be addresses in the discussion section.
The corrections of gene/protein annotation were not considered: example the Gallus gallus Cyp2D6 gene encodes the proteinCYP2D49 etc.
Author Response
To Reviewer.
Thank you very much for reviewing our manuscript again. We also greatly appreciate the reviewer for your precious comments and suggestions. Your delicate and detailed comments and suggestions have improved the quality of our manuscript, and all authors have also deepened their knowledge of research and writing manuscript. We hope that our edits and responses we provide below satisfactorily address all the comments you have noted.
Reviewer 1
Comments 1: The authors have answered to all questions addressed by this referee but failed to amend the manuscript accordingly (only very minor editorial changes were made). The introduction does not reflect the outcome of previous study, (See Kang, D.; Park, J.; Shim, K. Heat Treatment at an Early Age Has Effects on the Resistance to Chronic Heat Stress on Broilers. Animals, 2019, 9, 1022 – Journal missing in the ref list).
Answer 1: Thank you for your comments. We have changed your recommended sentence in introduction and corrected the reference name in reference section.
Comments 2: A major concern is the Discussion Section presenting a (text-book based) description of the genes/proteins under consideration but fails to establish a clear link to the effect of heat stress in poultry. Typical examples for the no-critical selection of literature information on metabolic pathways are the citation of references 9, 12, 16, 17, 18, 19, 21, 22, 24, 27, 29, 31, 32 which cover enterally different fields and disease condition.
Answer 2: Thank you for your inquiry. We skipped some reference following your inquiry throughout the manuscript.
Comments 3: It is strongly recommended that the authors discuss these results, with a competent colleague with broad insight into poultry physiology and pathophysiology and link the current results to the previous manuscript (Kang, D.; Park, J.; Shim, K. Heat Treatment at an Early Age Has Effects on the Resistance to Chronic Heat Stress on Broilers. 2019, 9, 1022).
Answer 3: Thank you for your recommendation. We have discussed with a competent colleague, and he was corrected this current manuscript. And we added it to the acknowledgment, thanks for the help of our colleagues.
Specific comments repeated:
Comments 1: The Short abstract is still inadequate as it does to provide easy understandable information about the content of the manuscript.
Answer 1: Thank you for your observation. We have rewrite the short abstract. We hope this abstract is now easily understandable to reader.
Comments 2: The authors should consider phrases like: based on our previous results, it was hypothesized that heat stress early in life may increase the resilience to heat stress at later stages of life. Hence in the current series of experiments chickens were exposed….
Answer 2: Based on reviewer suggestion we have change the recommended phrase in introduction part.
Comments 3: The recommended editorial changes in the figures were not implemented.
Answer 3: Thank you for your observation. We are apologies for our mistake. We have change the figure according to editorial comments.
Comments 4: The recommendation to present a clear link with clinical data (previous experiments) was not considered (only a statement was given in the Rebuttal letter).
Answer 4: The statement of clinical data of our current study was wrote in only the Rebuttal letter. However, this statement we have written in sampling subsection of Materials and method section.
Comments 5: Gene expression differences were presented in percent (without standard deviation. see figure 3) while the general procedure is to present x-fold changes. As none of the changes apparently resulted in >2 fold differences in the level of expression, it seems surprising that major changes in protein expression levels were observed. This apparent discrepancy should be addresses in the discussion section.
Answer 5: Figure 3 does not explain the differences in gene expression, but shows ‘the number of genes that encoded differentially expressed proteins included in each GO term/the total number of genes in GO term’. And Figure 4 shows gene expression differences in fold-changes compared to CC. Gene expression is generally used to verify the level of protein expression, however, gene expression is not necessarily the same as that of a protein level. Because the heat or other cell stress could change the levels of gene included transcription, mRNA processing, and translation etc. In order to verify protein expression more accurately, western blot or ELISA should have been used, but because we could not obtain a suitable antibody and other equipment due to our laboratory conditions, we wanted to verify it through mRNA expression.
- Holcik, M., & Sonenberg, N. (2005). Translational control in stress and apoptosis. Nature reviews Molecular cell biology, 6(4), 318-327.
- Gibson, G. (2008). The environmental contribution to gene expression profiles. Nature reviews genetics, 9(8), 575-581.
- Biamonti, G., & Caceres, J. F. (2009). Cellular stress and RNA splicing. Trends in biochemical sciences, 34(3), 146-153.
Comments 6: The corrections of gene/protein annotation were not considered: example the Gallus gallus Cyp2D6 gene encodes the protein CYP2D49 etc.
Answer 6: The Cyp2D6 is cytochrome P450 family 2 subfamily D member 6. The CYP2D49 is the synonyms of Cyp2D6. The symbol of protein and gene is sample. Therefore, we used only one symbol (Cyp2D6) for chicken. We attached some screenshot from NCBI for more conformation. And we changed the annotation about ‘name’ in Supplementary file to ‘gene’.

Reviewer 2 Report
In my opinion this corrected manuscript can be published.
Author Response
To Reviewer.
Thank you again very much for reviewing our manuscript. Your delicate and detailed comments and suggestions have improved the quality of our manuscript, and all authors have also deepened their knowledge of research and writing manuscript.